# Differential microRNA Expression in *USP8*-Mutated and Wild-Type Corticotroph Pituitary Tumors Reflect the Difference in Protein Ubiquitination Processes

**DOI:** 10.3390/jcm10030375

**Published:** 2021-01-20

**Authors:** Mateusz Bujko, Paulina Kober, Joanna Boresowicz, Natalia Rusetska, Natalia Zeber-Lubecka, Agnieszka Paziewska, Monika Pekul, Grzegorz Zielinski, Andrzej Styk, Jacek Kunicki, Jerzy Ostrowski, Janusz A Siedlecki, Maria Maksymowicz

**Affiliations:** 1Department of Molecular and Translational Oncology, Maria Sklodowska-Curie National Research Institute of Oncology, 02-781 Warsaw, Poland; mateusz.bujko@coi.pl (M.B.); paulina.kober@gmail.com (P.K.); joanna.boresowicz@gmail.com (J.B.); natarusetska@gmail.com (N.R.); jas@coi.waw.pl (J.A.S.); 2Department of Gastroenterology, Hepatology and Clinical Oncology, Medical Centre for Postgraduate Education, 01-813 Warsaw, Poland; natalia.zeber@o2.pl (N.Z.-L.); jostrow@warman.com.pl (J.O.); 3Department of Neuroendocrinology, Centre of Postgraduate Medical Education, 01-813 Warsaw, Poland; agapaziewska@poczta.onet.pl; 4Department of Pathology and Laboratory Diagnostics, Maria Sklodowska-Curie Institute-Oncology Center, 02-781 Warsaw, Poland; monika.pekul@pib-nio.pl; 5Department of Neurosurgery, Military Institute of Medicine, 04-141 Warsaw, Poland; gzielinski@wim.mil.pl (G.Z.); astyk@wim.mil.pl (A.S.); 6Department of Neurosurgery, Maria Sklodowska-Curie Institute-Oncology Center, 02-781 Warsaw, Poland; jkunickii@gmail.com; 7Department of Genetics, Maria Sklodowska-Curie Institute-Oncology Center, 02-781 Warsaw, Poland

**Keywords:** Cushing’s disease, corticotroph PitNET, miRNA expression, gene expression, next generation sequencing, *USP8*, *USP48*, mutation

## Abstract

Background: *USP8* mutations are the most common driver changes in corticotroph pituitary tumors. They have direct effect on cells’ proteome through disturbance of ubiquitination process and also influence gene expression. The aim of this study was to compare microRNA profiles in *USP8*-mutated and wild-type tumors and determine the probable role of differential microRNA expression by integrative microRNA and mRNA analysis. Methods: Patients with Cushing’s disease (*n* = 28) and silent corticotroph tumors (*n* = 20) were included. *USP8* mutations were identified with Sanger sequencing. MicroRNA and gene expression was determined with next-generation sequencing. Results: *USP8*-mutated patients with Cushing’s disease showed higher rate of clinical remission and trend towards lower tumor volume than wild-type patients. Comparison of microRNA profiles of *USP8*-mutated and wild-type tumors revealed 68 differentially expressed microRNAs. Their target genes were determined by in silico prediction and microRNA/mRNA correlation analysis. GeneSet Enrichment analysis of putative targets showed that the most significantly overrepresented genes are involved in protein ubiquitination-related processes. Only few microRNAs influence the expression of genes differentially expressed between *USP8*-mutated and wild-type tumors. Conclusions: Differences in microRNA expression in corticotropinomas stratified according to *USP8* status reflect disturbed ubiquitination processes, but do not correspond to differences in gene expression between these tumors.

## 1. Introduction

Pituitary neuroendocrine tumors (PitNETs) represent about 10–20% of all intracranial neoplasms in adults. They may arise from various kinds of secretory cells of pituitary gland, including corticotroph cells, which produce adrenocorticotropic hormone (ACTH). Corticotroph PitNETs commonly cause ACTH-dependent Cushing’s disease (CD); however, a notable proportion of tumors originated from corticotropic pituitary cells are endocrinologically non-functioning and classified as silent corticotroph tumors commonly referred to as silent corticotroph adenomas (SCA). Both active and silent corticotroph PitNETs share a similar molecular profile [1,2].

Recently, notable progress in the understanding of pathogenesis of CD has been made [3], including the discovery of recurrent *USP8* mutations [4,5,6]. These mutations are observed in approximately 30–40% of patients suffering from Cushing’s disease as well as in silent tumors [4,6,7,8,9]. Patients with Cushing’s disease with and without *USP8* mutations have a slightly different clinical profile according to previously published data [4,6,8,10,11,12,13,14,15]. The studies showed that *USP8* mutation is related to lower tumor size [4,5,8] and clinical remission after surgery [8] [12]. Additionally, differences in expression of possible molecular predictive markers as *MGMT* or somatostatin receptor were also observed in this group of patients [2,8]. Perhaps testing the *USP8* mutation status in patients could provide some kind of clinically useful information; however, the clinical results published up to today are insufficient, and molecular consequences of this mutation are only partially recognized.

*USP8* gene encodes for deubiquitinase enzyme involved in the regulation of proteasomal protein degradation. *USP8* mutations are small, single codon deletions or missense variants that occur in the region involved in binding 14-3-3 proteins family members. Thus, these changes impair interactions between USP8 and 14-3-3 proteins, which normally suppress deubiquitinase activity [16]. As a result of mutation, USP8 activity is enhanced and leads to preventing proteasomal degradation of particular proteins and dysregulation of natural protein turnover. This was clearly shown in in vitro experiments, providing the explanation of the sustained EGFR signaling in *USP8*-mutated corticotroph PitNETs [4,5]. Since USP8 deubiquitinase has many molecular substrates, its impaired functioning has potentially a wide effect on the protein level. The pleiotropic effect of the mutation is reflected by differences in gene expression between *USP8*-mutated (*USP8*mut) and *USP8*-wild type (*USP8*wt) tumors [1,2]. Accordingly, differences in expression of particular proteins related to corticotroph tumors growth were also found [15]. The understanding of the biological difference between wild type and mutated tumors appears important, since *USP8* mutation may potentially serve as prognostic and predicting factor [2].

The aim of this study was to compare the profiles of microRNA (miRNA) expression in corticotroph tumors stratified according to *USP8* mutational status and to determine the potential role of differential miRNA expression. Moreover, mutations of another deubiquitinase-encoding gene (*USP48*) contribute to pathogenesis of some *USP8*wt tumors [17,18]. Mutations of both genes were determined and taken into account in data analysis.

## 2. Experimental Section

### 2.1. Patients and Samples

Pituitary tumor samples were collected during transsphenoidal surgery and fixed in formalin for routine diagnostic procedures, including immunohistochemical and ultrastructural evaluation. Archival formalin-fixed paraffin-embedded (FFPE) tissue samples from 48 patients, including 28 samples from patients with Cushing’s disease and 20 samples from patients with SCA from years 2013–2017, were included. Patients were diagnosed according to WHO criteria applied during the time of tissue sampling [19]. Diagnosis was based on results of immunohistochemical staining for pituitary hormones and Ki-67 labeling as well as commonly accepted ultrastructural features of corticotroph tumors [20]. For this study, all the tumor samples were reevaluated histopathologically by one pathologist to confirm the diagnosis and determine tumor tissue content within each sample.

The diagnosis of Cushing’s syndrome/hypercortisolism was based on standard hormonal criteria: increased urinary free cortisol (UFC) in three 24 h urine collections, disturbances of cortisol circadian rhythm, increased serum cortisol levels accompanied by increased or not suppressed plasma ACTH levels at 8 a.m., and a lack of suppression of serum cortisol levels to < 1.8 µg/dL during an overnight dexamethasone suppression test (1 mg at midnight). The pituitary etiology of Cushing’s disease was confirmed based on the serum cortisol levels or UFC suppression < 50% with a high-dose dexamethasone suppression test (2 mg q.i.d. (lat. quater in die = four time a day) for 48 h) or a positive result of a corticotrophin-releasing hormone stimulation test (100 mg i.v. (intravenously)) and positive pituitary magnetic resonance imaging. In the group of SCAs, none of the patients had any evidence of hypercortisolism based on clinical signs and symptoms as well as basic laboratory tests. ACTH levels were assessed using IRMA (immunoradiometric assay) (ELSA-ACTH, CIS Bio International, Gif-sur-Yvette Cedex, France). The analytical sensitivity was 2 pg/mL (reference range: 10–60 pg/mL). Serum cortisol concentrations were determined by the Elecsys 2010 electrochemiluminescence immunoassay (Roche Diagnostics, Mannheim, Germany). Analytical sensitivity of the assay was 0.02 μg/dL (reference range: 6.2–19.4 μg/dL). UFC was determined after extraction (liquid/liquid with dichloromethane) by electrochemiluminescence immunoassay (Elecsys 2010, Roche Diagnostics, Mannheim, Germany)—reference range: 4.3–176 μg/24 h. Bilateral inferior petrosal sinus sampling was used as a routine investigation tool in any patient with proven ACTH-dependent Cushing’s syndrome and negative or equivocal MRI findings (intrasellar lesion ≤ 6 mm) [21]). A macroadenoma was defined as a tumor with at least one diameter exceeding 10 mm, and the tumor volume was assessed with the diChiro Nelson formula (height × length × width × π/6). Invasive growth of the tumors was evaluated using Knosp grading [22]. Tumors with Knosp grades 0, 1 and 2 were considered non-invasive, while those with Knosp 3 and 4 were considered invasive.

Overall characteristics of the patients are presented in Table 1 while details are provided in Appendix A. The content of tumor tissue in each FFPE sample ranged between 80 and 100% (median 99%) (details in Appendix A). The study was approved by the local Ethics Committee of Maria Sklodowska-Curie Institute—Oncology Center in Warsaw, Poland. Each patient provided informed consent for the use of tissue samples for scientific purposes.

DNA and total RNA from FFPE samples was isolated using RecoverAll™ Total Nucleic Acid Isolation Kit for FFPE (Thermo Fisher Scientific, Waltham, MA, USA) and measured using NanoDrop 2000 (Thermo Fisher Scientific, Waltham, MA, USA). The samples were stored at −70 °C.

### 2.2. Genomic Mutation Testing

The presence of point mutation at the *USP8* hotspot (exon 14) and *USP48* hotspot (exon 10) was determined using Sanger sequencing. DNA was PCR amplified with FastStart Taq DNA Polymerase (Roche Diagnostics, Mannheim, Germany) using the GeneAmp 9700 PCR system (Applied Biosystems, Foster City, CA, USA). The PCR product was purified using ExoSAP-IT (USA Affymetrix, Cleveland, OH, USA), labeled with BigDye Terminator v.3.1 (Applied Biosytems, Foster City, CA, USA) according to the manufacturer’s instructions and analyzed by capillary electrophoresis with the ABI PRISM 3300 Genetic Analyzer (Applied Biosystems, Foster City, CA, USA), as described previously [1]. The following sequences of PCR primers were used: 5′-TCCACCCCTCCAACTCATAA and 5′-CTGACAGATTCAGAGTAGAAACT for *USP8* mutation testing as well as 5′-GCCCGGCTAAAGAATAAACA and 5′-TGCCTGCTATAATCCTGGAAA for identification of *USP48* variants.

### 2.3. Determining miRNA Expression Profile with Next Generation Sequencing (NGS)

The quality of small RNA fractions was assessed using Agilent 2100 Bioanalyzer with Small RNA Kit chip (Agilent, Santa Clara, CA, USA) and measured with Qubit RNA HS Assay Kits (Thermo Fisher Scientific, Waltham, MA, USA). One µg of total RNA was used to sequencing library construction with an Ion Total RNA-Seq Kit v2 (Thermo Fisher Scientific), according to the manufacturer’s protocol. Ion Xpress™ RNA-Seq Barcode Kit, which allows for multiplexed sequencing, was used for hybridization and ligation of RNA adapters. RNA reverse transcription and subsequent cDNA purification and library size selection were performed using Nucleic Acid Binding Beads. cDNA was PCR-amplified, followed by DNA purification and size selection. Amount and size distribution of the amplified DNA was determined using Bioanalyzer 2100 using a High Sensitivity DNA Kit (Agilent, Santa Clara, CA, USA). The length of miRNA ligation products in barcoded libraries ranged between 94 and 114 bp. Template preparation for clonal amplification of up to four miRNA libraries at a concentration of 18 pM, and loading of the PI chip, were performed using Ion Chef instrument, with Ion PI™ Hi-Q™ Chef Kit (Thermo Fisher Scientific, Waltham, MA, USA). An Ion Proton sequencer (Thermo Fisher Scientific, Waltham, MA, USA) was used for sequencing. Unmapped bam files were converted into fastq files with a bamToFastq script from bedtools. Read mapping to known human miRNAs (according to miRBase release 22) and reads quantification were performed using miRDeep2.14. Data normalization and differential expression analysis were performed using DESeq2. Filtration for low-expression miRNAs and miRNAs genes with less than five sequencing reads in at least half of the samples were excluded. Fold change of expression (FC) calculated as ratio of the normalized read-count value in *USP8*-mutated and *USP8*-wt tumors was used as a measure of expression difference. Differentially expressed miRNAs were defined as those with |FC| > 2 and adjusted *p*-value < 0.05.

### 2.4. Gene Expression Profiling

Gene expression profiles were determined in 24 FFPE samples of corticotroph tumors by sequencing of amplicon-based library representing whole transcriptome, as described in detail previously [1]. Ion AmpliSeq™ Transcriptome Human Gene Expression Kit (Thermo Fisher Scientific, Waltham, MA, USA) was used for library preparation and semiconductive sequencing technology with Ion Proton instrument, PI chip, and the sequencing reagents included in Ion PI™ Hi-Q™ Sequencing 200 Kit according to the manufacturer’s instructions (Thermo Fisher Scientific, Waltham, MA, USA). Data processing was performed using Bioconductor packages in R environment as described [1]. Differentially expressed genes (DEGs) were defined as those with adjusted *p*-value < 0.05.

### 2.5. Prediction of miRNA–mRNA Interactions

Analysis of the interactions between miRNAs and mRNAs was applied to determine the possible functional role of miRNAs, which are differentially expressed in *USP8*mut and *USP8*wt tumors. We used both mRNA target prediction and correlation analysis of the expression levels of particular miRNAs and their predicted mRNA targets in corticotroph tumor samples. 

The MicroRNA Data Integration Portal (mirDIP) algorithm, which combines multiple sources for miRNA target prediction [23], was used for the identification of possible mRNA targets. Only mRNAs that were predicted as targets with a probability scored as VeryHigh, according to the mirDIP criterion, were taken into account and included into downstream analyses.

Then, the correlation between the expression levels of identified potentially interacting miRNAs and mRNAs was assessed using normalized read-count data from small RNA sequencing and matched data from gene expression profiling for the same tumor samples (GSE132982 dataset). Spearman rank correlation was calculated using data from expression profiling of 24 tumors in R environment. Unadjusted *p* < 0.01 was considered relevant for correlation results.

### 2.6. Statistical Analysis

Datasets of quantitative variables were tested for the normal distribution with Shapiro-Wilk test. Variables with normal distribution were analyzed with two-sided unpaired t-test, while a two-sided Mann–Whitney U-test was used when normal distribution was not verified. Exact Fisher’s test was used for the analysis of proportions. Significance threshold of α = 0.05 was adopted. For the identification of differentially expressed miRNAs and genes, *p*-values were adjusted for multiple testing with the Benjamini–Hochberg method. The Spearman correlation method was used for correlation analysis. Data were analyzed using GraphPad Prism 6.07 (GraphPad Software). Hierarchical clustering analysis was conducted with Cluster 3.0, and the results were visualized using TreeView 1.6 software (Stanford University School of Medicine, Stanford, CA, USA).

## 3. Results

### 3.1. USP8 and USP48 Mutations

The incidence of hotspot mutations in *USP8* and *USP48* was determined with Sanger sequencing in 48 patients. *USP8* mutations were identified in 15/48 (35.4%) patients. *USP48* mutations were identified in 2/48 (4.2%) patients; both were female patients suffering from Cushing’s disease with a diagnosis of densely granulated corticotroph tumors. One of the tumors was microadenoma while the other was macroadenoma. Both *USP48* mutations had p.Met415Ile substitution. *USP8* and *USP48* mutations were mutually exclusive. Details of the results are presented in Appendix A. The possible relationship between the incidence of *USP8* mutations and demographic/clinical parameters was investigated in groups of Cushing’s disease patients and SCA patients separately. Since only two patients with *USP48* mutation were identified, they were excluded from the analysis. No difference in age of diagnosis was observed between *USP8*-mutated (*USP8*mut) and *USP8*-wild type (*USP8*wt) patients, both in the Cushing’s disease group and in SCA group. Except for one male patient, all *USP8* mutations were identified in females; however, differences of proportions did not reach statistical significance in the Cushing’s disease group or SCA group.

All *USP8*mut patients suffering from Cushing’s disease were in clinical remission after surgery, while clinical remission was observed in 9/15 of *USP8*wt patients. Difference of proportions of patients with/without remission was significant (11/0 vs. 9/6; *p* = 0.0237). A trend towards lower tumor volume was observed in *USP8*mut patients vs. *USP8*wt patients in both the Cushing’s disease group (median 445 mm^3^ vs. 2730 mm^3^, respectively; *p* = 0.0798) and SCA group (median 1844 mm^3^ vs. 3893 mm^3^, respectively; *p* = 0.1707), but no significant difference was observed. Patients suffering from Cushing’s disease stratified according to *USP8* mutations status did not differ in terms of preoperative clinical parameters: morning serum ACTH level, morning serum cortisol level, or 24 h UFC.

Among patients with silent corticotroph tumors, significantly higher 24 h UFC level was observed in patients with *USP8* mutations than in *USP8*wt patients (median 124.4 vs. 66.32, respectively; *p* = 0.0334). No difference in morning serum ACTH level, morning serum cortisol level, or midnight serum cortisol level were observed between these patients. We did not observe any difference between *USP8*mut and *USP8*wt patients in invasive growth status as determined with Knosp grading, proliferation index, or histological subtype (sparsely vs. densely granulated) either in the group of Cushing’s disease patients or in those with silent tumors. The results are presented in Table 2.

### 3.2. Comparing miRNA Expression in USP8mut and USP8wt Corticotroph Tumors

The entire collection of 48 corticotroph tumors was subjected to miRNA expression profiling with next-generation sequencing of a small RNA fraction. Sequencing of small RNA libraries produced approximately 2,497,367 reads per sample, which were mapped to the human genome (hg19) and used for quantification of expression levels of known miRNAs, according to miRBase 22 release. Sequencing reads were annotated to 1917 miRNAs. Measurements of 1902 mature miRNAs were included in the analysis, after filtering out the miRNAs with low expression.

The overall analysis of the entire dataset with Principal Component Analysis (PCA) and hierarchical clustering methods did not show a clear separation between the groups of tumor samples stratified according to the mutation status, which indicates that the differences are not as pronounced as previously observed differences in gene expression profiles of *USP8*mut and *USP8*wt tumors [1,2]. Principal components 1 and 2 are presented in Appendix A, while a dendrogram showing similarity of miRNA expression between the samples is shown in Appendix A. Forty-six tumor samples were used for identification miRNAs differentially expressed in *USP8*mut and *USP8*wt tumors. Two samples with *USP48* mutations were excluded from differential analysis to avoid bias resulting from possibly different molecular features of these tumors. A total of 250 differentially expressed miRNAs were found (adjusted *p*-value < 0.05), including 68 miRNAs that met the criterion of |FC| > 2, as shown in Figure 1a,b. Most of them (57/68) were miRNAs with the expression higher in *USP8*mut than in wild-type tumors (Figure 1b).

### 3.3. Putative mRNA Targets for Differentially Expressed miRNAs

To identify the mRNA targets of 68 differentially expressed miRNAs, a two-step procedure was applied. First, miRNA–mRNA interactions were predicted with the use of an mirDIP tool [23], and subsequently, the correlation analysis of matched miRNA and gene expression profiles was applied. This analysis included 24 tumor samples. For 49 out of 68 miRNAs differentially expressed in *USP8*mut vs. *USP8*wt tumors, significant correlation with predicted target mRNA was observed. A total of 442 miRNA–mRNAs interactions were identified with a median of four putative target mRNAs per single miRNA particle (ranging from 1 to 38 target mRNAs). Mostly negative correlation between miRNA and gene expression was observed as found for 303 miRNA/mRNA pairs (range of Spearman R coefficient −0.575 to −0.7922; median −0.6182). Positive correlation was observed for 139 miRNA/mRNA pairs (range of Spearman R coefficient: 0.5753 to 0.8361 median: 0.6215). Results are presented in detail in Appendix A.

These analyses indicated 400 putative target genes that were identified as regulated by differentially expressed miRNAs. For the evaluation of potential functional significance of these genes, a subsequent gene set enrichment analysis (GSEA) was applied with the use of three gene ontology catalogs: KEGG Pathways, Gene Ontology (GO) Molecular Function and GO Biological Processes. Four KEGG pathways were found as significantly enriched for the putative target genes, including “Ubiquitin mediated proteolysis” as the most significantly enriched (according to *p*-value). The analysis with GO Molecular Function showed three protein ubiquitination related pathways as being in the top nine significantly enriched functions: Ubiquitin-like protein ligase activity (GO:0061659), Ubiquitin protein ligase activity (GO:0061630), and ubiquitin-protein transferase activity (GO:0004842). GO Biological process database indicates two processes related to the regulation of transcriptional activity and protein ubiquitination (GO:0016567) as the three most significantly enriched process (Figure 2a). The details of GSEA results are presented in Appendix A. The genes with a clear ubiquitination function, which are common for the ubiquitination-related processes and pathways, are listed in Table 3 with details of the miRNA/mRNA correlation analysis. Since a difference in interaction patterns between miRNAs with negative and positive miRNA-gene correlation were described previously [24], GSE analyses were also performed separately for putative gene targets, the expression of which is positively or negatively correlated with DEMs. This showed no significant overrepresentation for genes with positive miRNA–mRNA correlation, while we observed a clear enrichment of protein ubiquitination-related pathways and processes for genes characterized by negative miRNA–mRNA correlation. The analysis with GO Biological process indicated a proteasome-mediated ubiquitin-dependent protein catabolic process (GO:0043161), protein polyubiquitination (GO:0000209), and protein ubiquitination (GO:0016567), while GO Molecular Function showed ubiquitin-like protein ligase activity (GO:0061659), ubiquitin protein ligase activity (GO:0061630), ubiquitin-protein transferase activity (GO:0004842), RNA binding (GO:0003723), and glucocorticoid receptor binding (GO:0035259) as significantly enriched (presented in Figure 2b and in Appendix A).

### 3.4. Difference in miRNA Profile and Differential Gene Expression

The difference in gene expression profiles between *USP8*mut and *USP8*wt tumors was determined using sequencing data for 24 tumor samples, which were also included in miRNA–mRNA correlation analysis. The results of differential analysis indicated 1648 DEGs that met the criterion of adjusted *p* < 0.05. In order to identify DEGs with expression differences resulting from distinct miRNA profile in corticotroph tumors, stratified according to *USP8* mutation status, we compared the results of three analyses: differential gene expression analysis, differential miRNA expression analysis, and identification of putative miRNA–mRNA interaction. DEGs with a direction of expression fold change that is concordant with the sign of correlation and the expression change of the corresponding miRNA were considered relevant. In case of genes with a negative miRNA–mRNA correlation, we looked for genes with opposite fold change signs, while in the case of genes with positive correlation, we looked for genes with the same fold change signs.

Out of the 400 target mRNAs of which the expression correlated with differentially expressed miRNAs, 25 genes had significantly different expression level in *USP8*mut and *USP8*wt tumors. This included 21 genes with negative correlation of miRNA–mRNA expression: *KDM5A*, *KMT2C*, *SLAIN2, PGGT1B*, *RBM33*, *SNX13*, *KIAA0355*, *ANKRD52*, *PCDHAC2*, *CCDC88C*, *AFF1*, *GAS1*, *APLF*, *DNAJC6*, *RASAL2*, *LRP12*, *FAM135A*, *GMFB*, *SORT1*, *FAM133B* and *NFIA*. In turn, it includes four genes with positive expression correlation: *RAB15*, *PALM2*, *ELMO2*, and *JPH3*, which were differentially expressed. These 25 DEGs are putative targets of 12 miRNAs: hsa-miR-96-5p, hsa-miR-708-5p, hsa-miR-655-3p, hsa-miR-539-5p, hsa-miR-498, hsa-miR-382-5p, hsa-miR-383-5p, hsa-miR-330-3p, hsa-miR-329-3p, hsa-miR-326, hsa-miR-513a-5p, and hsa-miR-153-3p. The results are visualized in Figure 3 and presented in detail in Table 4.

## 4. Discussion

Hotspot mutations in the *USP8* gene, encoding ubiquitin carboxyl-terminal hydrolase 8, are the most common driver changes in corticotroph PitNETs. They have been detected in 30–40% of patients in previous studies [4,6,7,8], as also observed in our cohort.

The clinical relevance of *USP8* mutations was examined in few previous studies [4,6,8,10,11,12,13,14,15]. Mutations were identified predominantly in women [4,5,12,13,25] and more frequently in younger patients according to some reports [6,14]. We found mutations nearly exceptionally in women (with only one male patient); however, due to a high overrepresentation of female patients in the study group, we cannot conclude that there is a sex-related difference. We also did not observe a relationship between mutation and age of onset.

In patients with Cushing’s disease, these mutations appear related to lower tumor size and clinical remission after surgery [8,12], which could allow to consider *USP8* mutations a possible favorable prognosis marker. However, they are also related to a higher risk of recurrence [14]. In concordance with these previous observations, we found a significantly higher rate of clinical remission in *USP8*mut patients suffering from Cushing’s disease and a trend of lower tumor volume in mutated patients; however, no statistically significant difference was determined. We did not find differences in preoperative 24-h urinary free cortisol, which was observed to be significantly higher in patients with mutations in some previous studies [6,14]. Moreover, we did not find the difference in the proportion of tumors with invasive growth reported in other published results [5].

The analysis of the role of *USP8* mutations in silent corticotroph tumors has not been reported previously. The mutations are less frequent in this group of patients than in the case of Cushing’s disease. We identified the mutations in 20% of patients, while the mutation rate was 10% in the other study by Castellnoum, which included 20 SCAs [9]. No SCA with mutation was found in a study that included 11 such patients [6]. Because only four mutated SCAs were included in the analysis, the results may only be treated as preliminary. The only clinical parameter that we found significantly different in SCA patients with and without mutation was 24 h UFC, which was significantly higher in *USP8*mut patients; however, it was still within the reference range.

The generalization of our results should be done cautiously, and we have to emphasize an important limitation of our clinical data analysis. The numbers of patients included in the analysis are probably too low to draw a firm conclusion and the group is not representative of the general population, especially in the case of patients suffering from Cushing’s disease. Since the primary aim of our study was molecular profiling of tumor tissue, we intentionally preselected large tumors, which allowed us to have enough tissue for DNA/RNA isolation and successful molecular procedures.

From a molecular biology point of view, *USP8* mutations cause deregulation of the protein polyubiquitination/deubiquitination balance and impair the normal proteasomal degradation process [16]. In corticotroph tumors, a sustained EGFR signaling was found to be a consequence of the mutation [4,5] and the *USP8* changes probably affect normal turnover of many other proteins regulated by ubiquitin carboxyl-terminal hydrolase 8 [26]. The pleiotropic effect of the mutation is observed not only at the level of protein degradation, but it is also manifested at the level of gene expression [1,2]. In this study, we aimed to characterize the difference in miRNA profile in tumors with *USP8* mutations and wild-type PitNETs.

Recently, a novel driver mutation in another deubiquitinating enzyme i.e., *USP48*, was found in patients negative for *USP8* changes. *USP48* pathogenic variants cause increased activity of encoded deubiquitinase against its substrates Gli1 and H2A [17]. Because *USP48* mutations affect the processes of protein degradation similarly to mutations of *USP8*, we screened the tumor samples for both the mutations. Two samples with *USP48* Met/Val variant were identified and excluded from differential miRNA analysis to avoid a bias resulting from similarities in pathogenic mechanism. Due to a very low number of *USP48*-mutated tumors, these patients were also excluded from the analysis of clinical data.

We found that *USP8*mut and *USP8*wt tumors differ in miRNA expression; however, the differences are less pronounced than previously reported differences in mRNA expression [1,2]. We did not observe a clear distinction of *USP8*mut and *USP8*wt tumors in overall analysis, including PCA or hierarchical clustering based on the entire set of miRNA sequencing data, while a pronounced difference in mRNA expression profiles between *USP8*mut and *USP8*wt was reported previously [1,2].

*USP8*mut and *USP8*wt tumors differ in levels of relatively small proportion of all miRNAs that were included in differential analysis (approximately 3.5%). Conversely, a much higher proportion of differentially expressed protein-coding mRNAs was identified in corticotroph tumors stratified according *USP8* mutational status [1,2].

Since miRNAs play a regulatory role in expression and translation [27], the consequence of differential miRNAs’ expression depends on particular mRNA targets. To determine the possible functional role of DEMs in a high-throughput approach covering multiple miRNA–mRNA interactions, we predicted the putative mRNA targets for each DEM, followed by calculating the correlation coefficient for the expression levels of matched miRNAs and mRNAs. With this procedure, we mostly identified miRNA–mRNA pairs with negative correlation between expression levels where a high level of a particular miRNA corresponded with a low expression of its target gene. This relationship is concordant with a generally accepted concept that miRNAs are negative regulators of gene expression. Still, a notable part of the identified putative target mRNAs showed positive miRNA–mRNA correlation, indicating an activating role of miRNA. Activating action was previously reported for many miRNAs [28]. Recently published pan-cancer analyses [24,29] reported that many of miRNAs dysregulated in human cancer are positively correlated with their target genes.

GSE analysis was applied to identify the pathways where the identified target genes are overrepresented. The results showed that target genes of DEMs, especially those with negative miRNA–mRNA correlation, are related mainly to pathways and processes of protein ubiquitination. Since direct effects of *USP8* mutations are the changes at protein ubiquitination level [4,5,6], we believe that the different expression of miRNAs that are involved in editing ubiquitin marks may reflect this major biological difference between *USP8*mut and *USP8*wt corticotroph PitNETs. Protein ubiquitination processes are directly orchestrated by a high number of proteins belonging mainly to three classes: ubiquitin-activating enzymes (E1), ubiquitin conjugating enzymes (E2), and ubiquitin ligases (E3) [30]. The enzymes of each class catalyze the subsequent stages of protein ubiquitination and the reverse reaction of protein deubiquitination is conducted by deubiquitinating enzymes [30]. Each class of the enzymes involved in editing ubiquitin marks includes multiple proteins, and genes encoding for proteins belonging to each class were found as putative targets of DEMs.

Distinct expression of miRNAs appears to have a limited effect on differential gene expression in *USP8*mut and *USP8*wt tumors. Less than 10% of predicted target genes with correlated miRNA–mRNA expression levels have significantly different expression in corticotroph tumors with and without mutation. The mRNA level of only 25 out of over 1600 DEGs could be considered as related to a different miRNA expression. This means that factors other than miRNA are responsible for the previously described highly different gene expression profile in *USP8*mut and *USP8*wt PitNETs.

None of the ubiquitination processes-related genes that were identified as putative targets of DEMs have significantly distinct expression levels. However, some of DEGs that were identified as targets of DEMs may have an interesting role in the biology of *USP8*mut corticotroph PitNETs. For example, our results indicate that hsa-mir-382-5p, which has higher expression in *USP8*mut tumors, may regulate genes involved in transcriptional regulation: *KMT2C*, *KDM5A*, and *AFF1*. *KMT2C* encodes for lysine methyltransferase that introduces mono-methylation mark at histone H3K4 [31], while KDM5A is lysine demethylase that converts di- and trimethylated H3K4 into mono-methylated form [32]. H3K4 mono-methylation plays a role in regulation of gene enhancers activation [33]. In turn, AFF1 functions as a regulator of transcription elongation and chromatin remodeling [34]. This suggests that hsa-mir-382-5p may contribute to a large difference in gene expression levels between *USP8*mut and *USP8*wt corticotrophinomas.

Our results also suggest that hsa-miR-655-3p, which has higher expression in mutated tumors, may affect expression levels of *GAS1*. The protein encoded by this gene is a negative regulator of the cell cycle and is considered a tumor suppressor in gastric and colorectal cancer [35,36]. It is known that corticotroph PitNETs with a distinct *USP8* status differ in the expression status of cell cycle regulators at gene and protein level [1,2,15].

It is worth emphasizing that miRNA–mRNA interaction analysis results based on target prediction and calculation of expression correlation should be treated as preliminary. The commonly used methodology of detail validation of miRNA–mRNA interactions utilize laborious experimenting in vitro to confirm the impact of miRNA level on target gene expression and confirmation of physical miRNA–mRNA interaction with luciferase assay. This wet-lab approach is practically unfeasible for simultaneous investigation of many target genes of multiple miRNAs that we attempted to perform in our study. Additionally, this approach requires an appropriate cell model, but no human cell line of corticotroph cells is available. The only stable line of corticotroph cells are mouse AtT-20 cells and its usefulness in investigation of miRNA–mRNA interaction in human is limited due to evolutionary differences between species [37]. Some data on miRNA function in corticotroph cells based on mouse cell line were published [38,39,40]; however, it must be taken into account that approximately 46% miRNAs are considered primate-specific, while 14% are human-specific [37].

In summary, in our study we compared miRNA profiles of *USP8*mut and *USP8*wt corticotroph PitNETs and determined miRNAs with different expression levels. With target prediction and comprehensive miRNA and mRNA expression data analysis, we found that putative targets of DEMs are mainly the genes involved in processes and pathways of protein ubiquitination. However, differences in only a few miRNAs appear to affect the levels of genes with significantly diverse expression in corticotrophinomas with and without *USP8* mutations. Thus, the difference in miRNA levels is not the cause of a pronounced differences in the gene expression between these tumors.

## Figures and Tables

**Figure 1 jcm-10-00375-f001:**
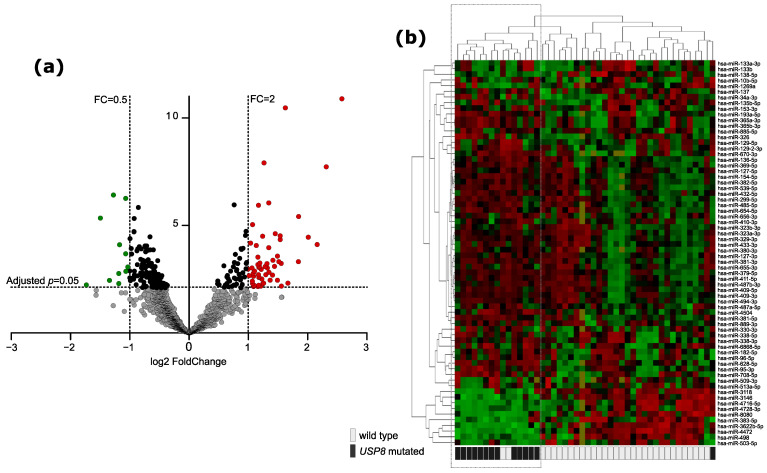
Difference in miRNA expression between *USP8*-mutated and *USP8* wild-type corticotroph PitNETs. (**a**) Volcano plot showing differentially expressed miRNAs. Significance and fold change thresholds are marked with dashed lines. (**b**) The expression levels of differentially expressed miRNAs in tumor samples stratified according to *USP8* mutation status with hierarchical clustering of the samples.

**Figure 2 jcm-10-00375-f002:**
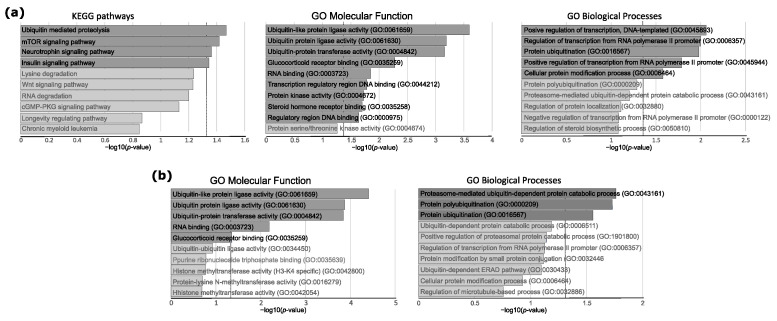
Gene Set Enrichment analysis of genes that were identified as regulated by miRNAs differentially expressed in *USP8*mut and *USP8*wt corticotroph tumors. (**a**) Results of the analysis of all putative target genes (*n* = 400). (**b**) Results for putative target genes with negative miRNA–mRNA correlation of expression levels (*n* = 239). Top 10 enriched pathways/processes are presented. Dark gray bars indicate significantly enriched pathways/processes (adjusted *p* < 0.05). Vertical dashed line indicates the significance threshold.

**Figure 3 jcm-10-00375-f003:**
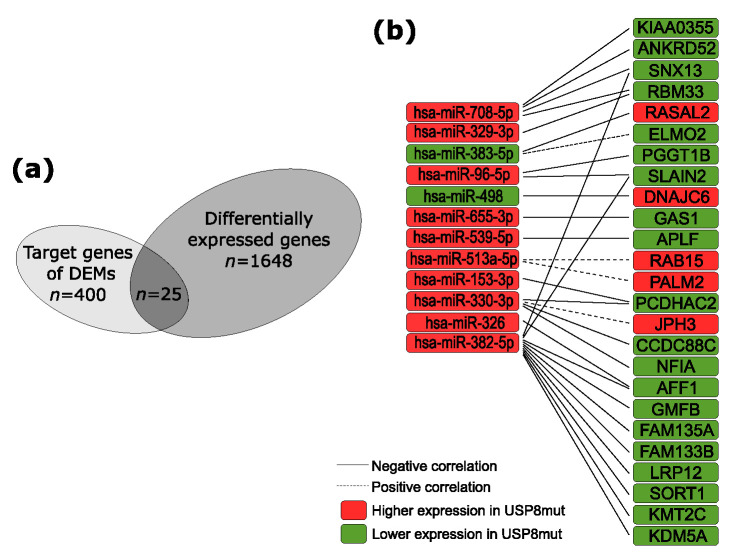
The predicted contribution of differentially expressed miRNAs (DEMs) in differential gene expression. (**a**) The proportion of genes differentially expressed in *USP8*mut and *USP8*wt corticotroph tumors; (**b**) genes that are putative targets of DEMs.

**Table 1 jcm-10-00375-t001:** Patients’ characteristics.

Number of Patients:	48
**Diagnosis (percentage of patients)**	
Cushing’s disease	58% (28/48)
Silent corticotroph adenoma	42% (20/48)
**Age (years)**	
Range	23–77
Median	49
**Gender (percentage of patients)**	
Male	25% (12/48)
Female	75% (36/48)
**Ultrastructural characteristics (percentage of patients)**	
Sparsely granulated	44% (21/48)
Densely granulated	56% (27/48)
**KNOSP grade (percentage of patients)**	
0	16% (7/48)
1	44% (21/48)
2	19% (9/48)
3	10% (5/48)
4	12.5% (6/48)
**Tumor size (percentage of patients)**	
Macroadenoma	77% (37/48)
Microadenoma	23% (11/48)
**Mutation status (percentage of patients)**	
*USP8* mutation	31% (15/48)
*USP48* mutation	4% (2/48)

**Table 2 jcm-10-00375-t002:** Summary of clinical features in patients with Cushing’s disease and silent corticotroph tumors regarding *USP8* mutation status.

Clinical Feature	Cushing’s Disease			Silent Corticotroph Tumors		
	*USP8*-mutated	*USP8*-wild type *	*p*-value	*USP8*-mutated	*USP8*-wild type	*p*-value
**Number of patients**	*n* = 11	*n* = 15		*n* = 4	*n* = 16	
**Sex** (ratio females/males)	10/1	13/2	1.00^a^	4/0	9/7	0.2487 ^a^
**Age at surgery** (years; median (range))	36 (23-67)	48 (24–76)	0.3993^b^	41.5 (23-77)	54.5 (34–77)	0.394 ^c^
**Cortisol 08:00 h** (µg/dL; median (range))	26.3 (21-49.7)	26.4 (11.9–38.6)	0.6036^c^	16.75 (9.1–50.8)	18.15 (6.8–29.7)	0.7408 ^c^
**ACTH 08:00 h** (pg/dL; median (range))	48.2 (37.3–102)	82.3 (36.9–129)	0.1945 ^c^	47.7 (42.1–61.3)	49 (14.7–74.9)	0.7408 ^c^
**UFC** (μg/24 h; median (range))	490 (276–810)	497 (215–739)	0.7974 ^b^	124 (94.76–139)	66.32 (13.70–126)	0.0334 ^c^
**Tumor volume** (mm^3^; median (range))	445.5 (32–6750)	2730 (62.5–6000)	0.0798 ^c^	1844 (900–7350)	3893 (1080–11088)	0.1707 ^c^
**Invasive tumor growth** (Knosp grade ratio 0, I, II/III, IV)	3/8	4/11	1.0000 ^a^	4/0	12/4	0.5377 ^a^
**Proliferation index** (ratio Ki67 ≥ 3%/Ki67 < 3%)	4/7	3/12	0.4065 ^a^	1/3	4/12	1.0000 ^a^
**Clinical remission**	11/0	9/6	0.0237 ^a^	-	-	-
**Histology** (ratio sparsely/densely granulated)	3/8	6/9	0.6828 ^a^	2/2	10/6	1.0000 ^a^

* Two patients with *USP48* mutations were excluded; ^a^ indicates the use of exact Fisher’s test; ^b^ indicates the use of a two-sided unpaired t-test ^c^ indicates the use of a Mann–Whitney U-test.

**Table 3 jcm-10-00375-t003:** The list of protein ubiquitination-related genes regulated by differentially expressed miRNAs (according to target prediction and correlation analysis) that were commonly found in the significantly enriched process in GSE analysis. The results of the correlation analysis between miRNA and predicted target mRNA levels and the results of miRNA differential analysis of *USP8*mut and *USP8*wt tumors.

Differentially Expressed miRNA	Fold Change of miRNA Expression	Adjusted *p*-Value	Predicted Target Gene	Spearman R	*p*-Value
			Ubiquitin-conjugating enzyme		
hsa-miR-182-5p	2.11	0.03245	*UBE2D4*	0.733	0.000359
hsa-miR-133a-3p	3.07	0.040107	*UBE2Q1*	0.720	0.000503
hsa-miR-153-3p	2.04	0.022436	*UBE2K*	−0.594	0.007305
hsa-miR-96-5p	2.09	0.023363	*UBE2K*	−0.582	0.008914
hsa-miR-137	2.57	0.003923	*UBE2G2*	−0.592	0.007544
hsa-miR-330-3p	2.62	0.00294	*UBE2J1*	−0.587	0.008197
			E3-ubiquitin ligase		
hsa-miR-137	2.57	0.003923	*RNF165*	−0.721	0.000495
hsa-miR-382-5p	2.71	0.022428	*KLHL42*	−0.704	0.000763
hsa-miR-433-3p	2.93	0.011477	*FBXO22*	−0.634	0.003579
hsa-miR-127-5p	2.14	0.022624	*PELI2*	−0.624	0.004318
hsa-miR-133b	2.79	0.029053	*KLHL9*	0.613	0.005292
hsa-miR-498	0.37	0.001367	*AMFR*	−0.603	0.006319
hsa-miR-329-3p	2.79	0.006727	*PELI2*	−0.592	0.007635
hsa-miR-338-5p	2.06	0.003694	*PJA2*	0.590	0.007876
hsa-miR-153-3p	2.04	0.022436	*RNF26*	−0.587	0.008254
hsa-miR-410-3p	2.65	0.011477	*RNF144B*	−0.585	0.008572
hsa-miR-498	0.37	0.001367	*MARCH4*	−0.583	0.008784
hsa-miR-432-5p	2.85	0.00294	*KLHL20*	−0.580	0.009173
hsa-miR-432-5p	2.85	0.00294	*CUL5*	−0.578	0.009531
			Deubiquitinase		
hsa-miR-381-3p	2.50	0.022163	*USP46*	−0.689	0.001117
hsa-miR-498	0.37	0.001367	*USP46*	−0.664	0.001953

**Table 4 jcm-10-00375-t004:** Genes with distinct expression levels in *USP8*mut and *USP8*wt tumors with an expression difference related to the levels of differentially expressed miRNAs.

		Correlation Analysis		Differential Gene Expression		Differential miRNA Expression	
Gene	MicroRNA	Spearman R	*p*-value	Fold change	Adjusted *p*-value	Fold change	Adjusted *p*-value
*PGGT1B*	hsa-miR-96-5p	−0.739	0.000302	0.55	0.015538	2.09	0.023363
*SLAIN2*	hsa-miR-96-5p	−0.689	0.001096	0.65	0.043609	2.09	0.023363
*RBM33*	hsa-miR-708-5p	−0.734	0.000348	0.70	0.024614	4.84	0.000013
*SNX13*	hsa-miR-708-5p	−0.700	0.000684	0.62	0.030738	4.84	0.000013
*KIAA0355*	hsa-miR-708-5p	−0.664	0.001928	0.59	0.033981	4.84	0.000013
*ANKRD52*	hsa-miR-708-5p	−0.617	0.004874	0.65	0.008329	4.84	0.000013
*GAS1*	hsa-miR-655-3p	−0.612	0.005347	0.27	0.020942	2.17	0.037144
*APLF*	hsa-miR-539-5p	−0.615	0.005108	0.15	0.003026	2.77	0.013880
*RAB15*	hsa-miR-513a-5p	0.601	0.006558	2.67	0.015333	2.11	0.006640
*PALM2*	hsa-miR-513a-5p	0.603	0.006263	2.34	0.032974	2.11	0.006640
*DNAJC6*	hsa-miR-498	−0.659	0.002133	1.65	0.0347725	0.37	0.001367
*RASAL2*	hsa-miR-383-5p	−0.611	0.005419	1.72	0.035180	0.46	0.048076
*ELMO2*	hsa-miR-383-5p	0.680	0.001352	0.54	0.002362	0.46	0.048076
*KDM5A*	hsa-miR-382-5p	−0.762	0.000149	0.69	0.013025	2.71	0.022428
*SNX13*	hsa-miR-382-5p	−0.729	0.000400	0.62	0.030738	2.71	0.022428
*KMT2C*	hsa-miR-382-5p	−0.701	0.000819	0.63	0.002765	2.71	0.022428
*LRP12*	hsa-miR-382-5p	−0.698	0.000880	0.61	0.013395	2.71	0.022428
*FAM135A*	hsa-miR-382-5p	−0.696	0.000923	0.70	0.042200	2.71	0.022428
*GMFB*	hsa-miR-382-5p	−0.695	0.000952	0.65	0.012156	2.71	0.022428
*SORT1*	hsa-miR-382-5p	−0.610	0.005543	0.64	0.017393	2.71	0.022428
*FAM133B*	hsa-miR-382-5p	−0.595	0.007159	0.42	0.014833	2.71	0.022428
*AFF1*	hsa-miR-382-5p	−0.584	0.008682	0.65	0.020921	2.71	0.022428
*SLAIN2*	hsa-miR-382-5p	−0.583	0.008789	0.65	0.043609	2.71	0.022428
*NFIA*	hsa-miR-330-3p	−0.615	0.005067	0.37	0.010137	2.62	0.002940
*PCDHAC2*	hsa-miR-330-3p	−0.587	0.008183	0.56	0.00004	2.62	0.002940
*JPH3*	hsa-miR-330-3p	0.580	0.008024	4.88	0.000058	2.62	0.002940
*CCDC88C*	hsa-miR-329-3p	−0.689	0.001105	0.28	0.023204	2.79	0.006727
*RBM33*	hsa-miR-329-3p	−0.594	0.007343	0.70	0.024614	2.79	0.006727
*AFF1*	hsa-miR-326	−0.662	0.001997	0.65	0.020921	2.31	0.000016
*PCDHAC2*	hsa-miR-153-3p	−0.583	0.008848	0.56	0.036190	2.04	0.022436

## Data Availability

Results of small RNA sequencing were submitted to Gene Expression Omnibus database.

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
