# Peer review of "Differential microRNA Expression in *USP8*-Mutated and Wild-Type Corticotroph Pituitary Tumors Reflect the Difference in Protein Ubiquitination Processes"

_jcm, 2021, doi:10.3390/jcm10030375_

Round 1
Reviewer 1 Report
Thank you for the opportunity to review this manuscript. The authors have retrospectively analyzed biopsy samples from 48 patients operated for pituitary tumour, 28 pat. with Cushings’s disease and 20 pat. with silent corticotrophic adenoma. Most of the tumours were macroadenoma (however not 77 of them as written in table 1 – this is probably wrong typing for 37? )- , however most of the macroadenomas must have been small with KNOSP grade less of 0-1 in 28 /48 patients.
The tissue samples were divided into USP8 (15) and USP48 (2) mutated vs WT-tumours.
The aim of the study is clear: To compare the profiles of microRNA expression according to USP8 mutation status.
Conclusion: No significant difference in micoRNA expression were found between the USP8 mutated /wt – tumours, thereby other mechanisms are responsible for the different gen-expression between USP8/WT tumours.
Even with a negative result these data should be published. The methodology and the data are presented clearly.
Comments:
The corticotrophic cell line is characterized by the presence of transcription factor T-PIT, was this examined in these tumours.
Table 1. The number of macroadenomas is obviously not 77.
Was there any difference in tumour size / KNOPS grade between USP8/wt?
Author Response
Reviewer 1
Comments:
Reviewer’s comment: The corticotrophic cell line is characterized by the presence of transcription factor T-PIT, was this examined in these tumours.
Reply: Patients’ samples were collected within years 2013-2017 before implementation of new WHO 2017 criteria. Patients were diagnosed according to WHO 2004 criteria [1]. For this reason staining against T-PIT was not performed.
Histopathological diagnosis was based on immunostaining with a panel of antibodies against pituitary hormones and Ki67 as well as examination with electron microscopy. All the tumors were ACTH positive and had clear ultrastructural features of corticotroph tumors: shape and size of secretory granules, presence of cytokeratin fibrils around cell nuclei in densely granulated type [2](Details of the results of immunostaining with pituitary hormones for each patient was introduced in supplementary table 1 with detailed patients’ description.
Reviewer’s comment: Table 1. The number of macroadenomas is obviously not 77.
Reply: This mistake has been corrected in revised manuscript.
Reviewer’s comment: Was there any difference in tumour size / KNOPS grade between USP8/wt?
Reply: The analysis of clinical data of patients stratified according to USP8 mutation status was performed for the revised manuscript. In general, patients with silent corticotroph tumors and patients with Cushing’s disease highly differ in basic clinical parameters like UFC or tumor size, as also clearly observed in our data. Therefore, these two categories of patients (with SCA and Cushing’s disease) were analyzed separately. Following the suggestion from reviewer 2 patients with USP48 mutations were excluded from analysis. Only two USP48-mutated tumors were identified, and this number is to low for most of statistical procedures (for example Mann-Whitney test that we mainly used).
The results of the analysis were described in results section 3.1 and Table 2 and commented in Discussion section in the revised manuscript. Description of methodology of the analysis was introduced in section 2.6 Statistical analysis.
- DeLellis, R. a Pathology and genetics of tumours of endocrine organs. In World Health Organization classification of tumours.; 2004; p. 230 ISBN 9283224167.
- Asa, S.L. Tumors of the Pituitary Gland; American Registry of Pathology Press: Washington, 2011;
Reviewer 2 Report
List of comments:
The authors examine a cross-sectional of miRNA profile of USP8 mutated and wild type tumours of 48 patients with corticotropic pituitary tumours. The study is of interest.
However, some highly significant problems in the design preclude interpretation as detailed below.
Major comments
The authors collected samples from 48 patients; including 28 samples from patients with Cushing’s disease (CD) and 20 samples from patients with silent corticotropic, adenoma (SCA)
Regarding the pituitary aetiology of Cushing Syndrome (CS).
The authors have to ensure the samples are CD
Most of CD are microadenomas, arbitrarily defined as being less than 10 mm, with a mean of approximately 6 mm. Bilateral inferior petrosal sinus sampling (BIPSS) is considered the gold standard to differentiate pituitary from ectopic CS in discordant result or MRI lesions < 6mm to ensure that it is not an incidental microadeoma.
- Did the authors have any discordant result???
- I checked in the supplemental data they have at least 4 patients with tumour less than 6mm BIPSS was performed in this cases?
This point have to be clarified, because the CD diagnose it is complex and to have valid results you have to accurately ensure your sample are tumour from CD.
In the same line, the SCA diagnose have to be further explained. In addition to the absence of hypercortisolism, (not hypercortisolemia ** as the authors write) the diagnoses was establish after pathologic examination ?, positive staining for adrenocorticotrophic hormone and other features?? citoqueratin ?? transcriptional factor T-Pit….?? Some part of the information it is in the supplemental data but the authors have to write this important characteristic in the Patients and samples section.
To sum up, I strong recommend the authors ensure that, this tumours come from Corticotroph lineage, and describe it in the text properly.
From a clinical point of view it is more interesting to compare SCA to CD which have totally different behaviour but there is neither comparison or comment on this extremely important hot topic. I strongly recommend that the authors make the effort to compare and describe ( EC and SCA) USP8’ mutations differences which will give much more value to the results.
The objective of the study it is not linked with possible clinical benefit. The lack of histological or radiographic features that accurately predict pituitary tumour clinical behaviour, let to welcome all new “possible biomarkers”
Finally, sample with USP48 mutations (only 2) should be excluded and change the abstract , the introduction etc accordingly. The authors some time excluded it for de analysis but in other they take it into account , this make all the data analysis more confuse and the “real” focus of the study it is USP8 mutated vs wild type.
Minor comments
- Introduction
- P2 L81 reevaluated histopathologically. Highlight this point, it is important the authors stress that the authors s carefully certify this sample were from a pathology point of view from corticotroph lineage.
- P2 L72-74 Consider to delate “Moreover, mutations of another deubiquitinase-encoding gene (USP48)….
- Experimental Section
- P2 line 79. FFPE should first be spelled out
- P 2 line 86-89 : clarify the diagnostics test pointing the BIPSS out, the large of the tumours etc
- Table 1: The units should be written are (%) are crude number. I am not able to understand for example macroadenoma 77?? Microadenoma 11??
- P4 Line 138: adjusted in which variable?? USP8??
- P4 line 153: MirDIP: should first be spelled out
- Results
- P5 line 165-166, this part it is for YOUR results I cannot understand why the authors mention reference [1], if the authors want to clarify more the methods this is not the appropriate section
- P5 L178: PCA should first be spelled out
- P5 L 180 (Supplementary Figure 1) is not available!
- P5 L183 The authors say Two samples with USP48 mutations were excluded from differential analysis. nevertheless, in the paragraph line 187-188 the authors mention it , also in Figure 1 in the footnote
- P5 L 187 adjusted p-value: adjusted for which variable??
- P6 line 214 GO: spelled out
- Discusion
- Should be rewritten linking the results with clinical applications and further explain the results when are different from what is published from other authors, adding explanations for such differences

Author Response
Reviewer 2
Major comments
Reviewer’s comment: The authors collected samples from 48 patients; including 28 samples from patients with Cushing’s disease (CD) and 20 samples from patients with silent corticotropic, adenoma (SCA)
Regarding the pituitary aetiology of Cushing Syndrome (CS).
The authors have to ensure the samples are CD
Most of CD are microadenomas, arbitrarily defined as being less than 10 mm, with a mean of approximately 6 mm. Bilateral inferior petrosal sinus sampling (BIPSS) is considered the gold standard to differentiate pituitary from ectopic CS in discordant result or MRI lesions < 6mm to ensure that it is not an incidental microadeoma.
- Did the authors have any discordant result???
- I checked in the supplemental data they have at least 4 patients with tumour less than 6mm BIPSS was performed in this cases?
This point have to be clarified, because the CD diagnose it is complex and to have valid results you have to accurately ensure your sample are tumour from CD.
Reply: We used bilateral inferior petrosal sinus sampling as a routine investigation tool in any patient with proven ACTH-dependent Cushing’s syndrome and negative or equivocal MRI findings (intrasellar lesion ≤ 6 mm). BIPSS procedure was performed for the 4 patients with lesions < 6mm This information was introduced in the revised manuscript section 2.1. Patients and samples. No patients with any equivocal diagnosis was included.
We used term hypercortisolism instead of hypercortisolemia following the recommendation.
Reviewer’s comment: In the same line, the SCA diagnose have to be further explained. In addition to the absence of hypercortisolism, (not hypercortisolemia ** as the authors write) the diagnoses was establish after pathologic examination ?, positive staining for adrenocorticotrophic hormone and other features?? citoqueratin ?? transcriptional factor T-Pit….?? Some part of the information it is in the supplemental data but the authors have to write this important characteristic in the Patients and samples section.
Reply: Patients’ samples were collected within years 2013-2017 before implementation of new WHO criteria from 2017. Patients were diagnosed according to WHO 2004 criteria [1]. For this reason staining against T-PIT was not performed.
Diagnosis was based on immunostaining with a panel of antibodies against pituitary hormones and Ki67 as well as examination with electron microscopy. All the tumors were ACTH positive and had clear ultrastructural features of corticotroph tumors: shape and size of secretory granules, presence of cytokeratin fibrils around cell nuclei in densely granulated type [2]. Details of the results of immunostaining with pituitary hormones for each patient was introduced in supplementary table 1 with detailed patients description.
More detailed description of histopathological diagnosis was introduced in revised manuscript in section 2.1. Patients and samples.
Reviewer’s comment: To sum up, I strong recommend the authors ensure that, this tumours come from Corticotroph lineage, and describe it in the text properly.
Reply: More information on patients diagnosis was introduced in revised manuscript (section 2.1. Patients and samples.) and more details on diagnostic results for each patient are provided in revised supplementary table 1 with detailed description of patients.
Reviewer’s comment: From a clinical point of view it is more interesting to compare SCA to CD which have totally different behaviour but there is neither comparison or comment on this extremely important hot topic. I strongly recommend that the authors make the effort to compare and describe ( EC and SCA) USP8’ mutations differences which will give much more value to the results.
Reply: We agree that additional clinical analysis should be included in our manuscript.
As indicated by Reviewer, patients with silent corticotroph tumors and patients with Cushing’s disease highly differ in basic clinical parameters like UFC or tumor size, as also clearly observed in our data. Therefore, these two categories of patients (with SCA and Cushing’s disease) were analyzed separately.
Reviewer’s comment: The objective of the study it is not linked with possible clinical benefit. The lack of histological or radiographic features that accurately predict pituitary tumour clinical behaviour, let to welcome all new “possible biomarkers”
Reply: The analysis of clinical data of patients stratified according to USP8 mutation status was performed for the revised manuscript. It showed that tumors with USP8 mutation relates to clinical remission after surgery and tend to be smaller in size. No relationship between USP8 mutation and Knosp grade was found. This was were described in section Results 3.1 and presented in Table 2 as well as commented in Discussion section in the revised manuscript. Description of methodology of the analysis was introduced in section 2.6 Statistical analysis, which was included in the revised manuscript.
Reviewer’s comment: Finally, sample with USP48 mutations (only 2) should be excluded and change the abstract , the introduction etc accordingly. The authors some time excluded it for de analysis but in other they take it into account , this make all the data analysis more confuse and the “real” focus of the study it is USP8 mutated vs wild type.
Reply: Following this suggestion patients with USP48 mutations were excluded from all the analyses. We must agree that this low number of samples exclude any conclusive analysis (for example Mann-Whitney test requires at least 3 samples) and description of the results for these patients is confusing.
Minor comments
Introduction
Reviewer’s comment: P2 L81 reevaluated histopathologically. Highlight this point, it is important the authors stress that the authors s carefully certify this sample were from a pathology point of view from corticotroph lineage.
Reply: We mean that all tissue samples selected for this study (that were collected through the years) were histopathologically examined by one pathologist to confirm diagnosis. The aim of this assessment was also the verification of high content of tumor tissue within the samples which is important for molecular analysis. This information was more clearly provided in revised manuscript section 2.1. Patients and samples. Tumor tissue content is provided in Supplementary Table 1.
Reviewer’s comment: P2 L72-74 Consider to delate “Moreover, mutations of another deubiquitinase-encoding gene (USP48)….
Reply: We would rather leave this basic information. It was introduced to mark that both mutations cause changes in the same category of molecular pathways. This is why we determined the incidence of these mutations in the samples not include in molecular analysis USP48 mutated samples.
Experimental Section
Reviewer’s comment: P2 line 79. FFPE should first be spelled out
Reply: This has been corrected in revised manuscript.
Reviewer’s comment: Table 1: The units should be written are (%) are crude number. I am not able to understand for example macroadenoma 77?? Microadenoma 11??
Reply: This has been corrected in revised manuscript. Percentage of patients are provided in revised Table 1. The number 77 was a mistake, it should be 37.
Reviewer’s comment: P4 Line 138: adjusted in which variable?? USP8??
Reply: For the identification of differentially expressed miRNAs and differentially expressed genes with the use of sequencing results P-value was adjusted for multiple testing with Benjamini-Hochberg method. This information was introduced in Section 2.5 Statistical analysis in the revised manuscript.
Reviewer’s comment: P4 line 153: MirDIP: should first be spelled out
Reply: This was corrected in revised manuscript: MicroRNA Data Integration Portal (mirDIP)
Results
Reviewer’s comment: P5 line 165-166, this part it is for YOUR results I cannot understand why the authors mention reference [1], if the authors want to clarify more the methods this is not the appropriate section
Reply: The reference was moved to methods sections.
Reviewer’s comment: P5 L178: PCA should first be spelled out
Reply: This has been corrected in revised manuscript: Principal Component Analysis (PCA)
Reviewer’s comment: P5 L 180 (Supplementary Figure 1) is not available!
Reply: This was mistake. We uploaded Supplementary Figure 1 with revised manuscript.
Reviewer’s comment: P5 L183 The authors say Two samples with USP48 mutations were excluded from differential analysis. nevertheless, in the paragraph line 187-188 the authors mention it , also in Figure 1 in the footnote
Reply: Samples with USP48 mutations were excluded from identification of differentially expressed miRNAs but were included for visualization of the expression of the identified miRNAs in the entire collection of the samples. We agree this is confusing and following the general suggestion we excluded USP48 mutated samples from all the analysis. Figure 1 was corrected to exclude USP48 mutated samples
Reviewer’s comment: P5 L 187 adjusted p-value: adjusted for which variable??
Reply: For the identification of differentially expressed miRNA and differentially expressed genes with the use of sequencing results P-value was adjusted for multiple testing with Benjamini-Hochberg method. This information was introduced in Section 2.5 Statistical analysis in the revised manuscript.
Reviewer’s comment: P6 line 214 GO: spelled out
Reply: This has been corrected in revised manuscript: Gene Ontology (GO)
Discusion
Reviewer’s comment: Should be rewritten linking the results with clinical applications and further explain the results when are different from what is published from other authors, adding explanations for such differences
Reply: The paragraph to comment our results of clinical data analysis in USP8 mutation and wild type patients were introduced in Discussion section. Our results were compared to those previously published and original articles have been referenced. The statement regarding limitations of or results (low patients number and overrepresentation of large tumors) was also introduced in revised discussion.
- DeLellis, R. a Pathology and genetics of tumours of endocrine organs. In World Health Organization classification of tumours.; 2004; p. 230 ISBN 9283224167.
- Asa, S.L. Tumors of the Pituitary Gland; American Registry of Pathology Press: Washington, 2011;
Round 2
Reviewer 2 Report
The authors have made an effort at revising their manuscript, which is improved. However, there are still serious background problems that should have been corrected in this stage and therefore I do not see possible their publication in JCM
I expose only a few that could help the authors
- Abstract : must be improved. Is not easily understood. It is important putting in the methodology section of the abstract , that you included Cushing’s disease and SCA samples.
- Supplementary Table1 it does not exist! you have uploaded a " figure 1" but it does not seem to be related to what it indicates
- Line: 217-220: writing problems
- Table 2 it is welcome ,but, you have some inaccuracies:
- Probably because of the low number of patients, your data not follow the normal distribution thus median and range are preferred.
- You express your results in mean and (SEM). SEM is not a descriptive statistics and should not be used as such, mean and SD or even better a 95% confidence interval ( if normal distribution)
- add in the foot note the statistical test for comparisons For ex. Fisher’s exact test for categorical variables and Mann–Whitney U-test for continuous variables.
Author Response
We thank the reviewer for the valuable comments. We also appreciate the prompt reply with reviewer's report. Below we would like to provide the replies for particular comments.
Reviewer’s comment: Abstract : must be improved. Is not easily understood. It is important putting in the methodology section of the abstract , that you included Cushing’s disease and SCA samples.
Reply: We made an attempt to improve the abstract. We put the information on number of patients with Cushing disease and SCAs in abstract (section methods). However, the limit is 200 words in the abstract including section headers, according to publisher’s recommendations, thus, it is not possible to include more detailed information.
Reviewer’s comment: Supplementary Table1 it does not exist! you have uploaded a " figure 1" but it does not seem to be related to what it indicates
Reply: Both supplementary files (Supplementary Tables and Supplementary figure were zipped and uploaded with revised manuscript. We can see the uploaded zipped supplementary files and download it through on-line manuscript submission system. We do not know why it is not available for reviewers. We will ask the editors to provide both supplementary files to the reviewers with this submission.
Supplementary figure 1 is referenced only once in the manuscript: “The overall analysis of the entire dataset with Principal Component Analysis (PCA) and hierarchical clustering methods did not show a clear separation between the groups of tumor samples stratified according to mutation status (Supplementary Figure 1) ….”
This supplementary figure presents two graphs: first showing two most important principal components (PC1 and PC2) for each tumor sample and the second graph is a dendrogram showing similarity of the samples in regard to overall miRNA expression which is a result of hierarchical clustering analysis. Both graphs represent the results mentioned in the main text. This was more clearly indicated in the main text in the revised manuscript.
Reviewer’s comment: Line: 217-220: writing problems
Reply: The sentence was corrected.
Reviewer’s comment: Table 2 it is welcome ,but, you have some inaccuracies:
- Probably because of the low number of patients, your data not follow the normal distribution thus median and range are preferred.
- You express your results in mean and (SEM). SEM is not a descriptive statistics and should not be used as such, mean and SD or even better a 95% confidence interval ( if normal distribution)
- add in the foot note the statistical test for comparisons For ex. Fisher’s exact test for categorical variables and Mann–Whitney U-test for continuous variables.
Reply: In fact most of our data do not follow normal distribution. In revised methods section/ 2.6 Statistical analysis we put the information that Shapiro-Wilk test was used for testing normal distribution and choice of the method of analysis was based on normal distribution status. In Table 2 in revised manuscript, two calculated p-values were changed (comparison of age and UFC in patients with Cushing disease). In these two cases normal distribution was verified and t-test could be used.
As suggested by the reviewer we presented median value and range for the variables detailed in Table 2 and when describing the results in Result section/3.1. The use of the particular statistical test is indicated in revised Table 2, as suggested.